# Nudging towards COVID-19 and influenza vaccination uptake in medically at-risk children: EPIC study protocol of randomised controlled trials in Australian paediatric outpatient clinics

Bing Wang,[1,2,3] Prabha Andraweera,[1,2,3] Margaret Danchin,[4,5,6] Christopher C Blyth,[7,8,9] Ivo Vlaev [ID],[10] Jason Ong,[11] Jodie M Dodd [ID],[2,3,12] Jennifer Couper,[2,3,13] Thomas R Sullivan [ID],[14,15] Jonathan Karnon,[16] Nicola Spurrier,[17,18] Michael Cusack,[17] Dylan Mordaunt,[18] Dimi Simatos,[19] Gustaaf Dekker,[2,3,20] Samantha Carlson,[7] Jane Tuckerman [ID],[4,6] Nicholas Wood,[21,22] Lisa J Whop,[23] Helen Marshall [ID][1,2,3]

**Correspondence to**
Professor Helen Marshall;
helen.marshall@adelaide.edu.au

## ABSTRACT

**Introduction** Children with chronic medical diseases are at an unacceptable risk of hospitalisation and death from influenza and SARS-CoV-2 infections. Over the past two decades, behavioural scientists have learnt how to design non-coercive 'nudge' interventions to encourage positive health behaviours. Our study aims to evaluate the impact of multicomponent nudge interventions on the uptake of COVID-19 and influenza vaccines in medically at-risk children.

**Methods and analyses** Two separate randomised controlled trials (RCTs), each with 1038 children, will enrol a total of approximately 2076 children with chronic medical conditions who are attending tertiary hospitals in South Australia, Western Australia and Victoria. Participants will be randomly assigned (1:1) to the standard care or intervention group. The nudge intervention in each RCT will consist of three text message reminders with four behavioural nudges including (1) social norm messages, (2) different messengers through links to short educational videos from a paediatrician, medically at-risk child and parent and nurse, (3) a pledge to have their child or themselves vaccinated and (4) information salience through links to the current guidelines and vaccine safety information. The primary outcome is the proportion of medically at-risk children who receive at least one dose of vaccine within 3 months of randomisation. Logistic regression analysis will be performed to determine the effect of the intervention on the probability of vaccination uptake.

**Ethics and dissemination** The protocol and study documents have been reviewed and approved by the Women's and Children's Health Network Human Research Ethics Committee (HREC/22/WCHN/2022/00082). The results will be published via peer-reviewed journals and presented at scientific meetings and public forums.

## STRENGTHS AND LIMITATIONS OF THIS STUDY

⇒ The Nudgeathon involved collaboration with various stakeholders who codesigned the nudge interventions.
⇒ Nudge interventions are currently undergoing evaluation in randomised controlled trials, with an additional assessment of cost-effectiveness.
⇒ The waiver of consent in our study helps mitigate selection bias.
⇒ The general public has experienced multiple COVID-19 vaccine campaigns, which may lead to fatigue with any further COVID-19 promotion strategies.

**Trial registration number** NCT05613751.

## INTRODUCTION

Children with chronic medical conditions represent a clinically vulnerable group in Australia with around 15% of children having either a respiratory, cardiac or neurological condition that increases their risk of hospitalisation and death from respiratory infections.[1] A systematic review and meta-analysis conducted by our group in 2017 demonstrates that compared with healthy peers, medically at-risk children with influenza are at higher risks for intensive care unit (ICU) admission, mechanical ventilation and death.[2] They are also more likely to develop bacterial pneumonia or experience prolonged hospital length of stay.[2] Australian data on children≤16 years admitted with

acute respiratory illnesses and tested for influenza at sentinel hospitals between 2010 and 2019 demonstrate that comorbidities are an independent predictor of severe outcomes. Specifically, the odds of ICU admission are higher in those with any comorbidity (adjusted OR (aOR): 1.36, 95% CI: 1.05 to 1.77) compared with cases without any comorbidity. Children with diabetes (aOR: 3.22, 95% CI: 1.25 to 8.23), cardiac (aOR: 1.93, 95% CI: 1.23 to 3.03), respiratory (aOR: 1.54, 95% CI: 1.08 to 2.21) or neurological comorbidities (aOR: 1.57, 95% CI: 1.25 to 1.98) are at greatest odds of ICU admission. Respiratory, neurological, cardiac, genetic and hepatic comorbidities and diabetes were shown to be associated with prolonged hospitalisation, whereas respiratory, renal, cardiac and other comorbidities increased the length of ICU stay.[3] The cost per episode of influenza-associated hospitalisation was estimated to be $A19 704 (95% CI: 11 715 to 27 693) for children with chronic lung conditions compared with $A4557 (95% CI: 4129 to 4984) for children without.[4] Influenza vaccines are provided free to this group (≥6 months).[5] Our research has previously shown that parents of children with chronic medical conditions are less likely to have their children vaccinated unless recommended by their specialist.[6] The uptake of influenza vaccine at least once in the last 2 years among medically at-risk children is 50% with annual vaccination only 32.8%.[7]

The SARS-CoV-2 infection causing COVID-19 may overall be milder in children compared with adults but a small proportion develop severe disease and are at-risk of a debilitating multisystem-inflammatory syndrome and/or long COVID (long lasting complications of COVID-19).[8] Children with chronic diseases are also at a higher risk of COVID-19 complications including pneumonia and respiratory failure and long-term health consequences.[4 9] Long COVID is a multisystem illness characterised by ongoing persistent symptoms that can last for weeks or months following SARS-CoV-2 infection.[10] Chronic fatigue, short-term memory problems, loss of taste and smell and long school absences were reported in long COVID cases.[11] COVID-19 vaccines are provided free to medically at-risk children (≥6 months).[5] Of concern, two-dose COVID-19 vaccine uptake among children aged 5–11 years is at present approximately 40% nationally.[12] At a time when Australia is still experiencing a large number of SARS-CoV-2 infections, with greater risks to the individual and the health system of cocirculation of influenza and SARS-CoV-2 infections and emergence of new viral variants, it is critical that all efforts are made to improve vaccine uptake among these high-risk groups. Equally important is the opportunity to continue to evaluate the safety of COVID-19 vaccines in these groups including children with chronic conditions that are often excluded from clinical trials.

Lack of access and practical barriers to vaccines remain the leading cause of low vaccine uptake, which is especially relevant to patients whose care is primarily in a hospital setting. For children with chronic conditions, the hospital often takes on the role of primary as well as tertiary care due to parental anxiety, life pressures, timeliness and access issues. Parents have identified barriers such as concern and lack of confidence in family physicians providing a vaccine recommendation for medically at-risk children and access to vaccines in hospital settings being suboptimal. Mobility, transport, language, financial barriers and remoteness are also reported barriers for children with chronic and complex conditions in accessing services.[13] For these groups, vaccination in the hospital setting, when patients are already onsite and engaged with a healthcare setting, would improve immunisation opportunities and protection against the threat of both influenza and SARS-CoV-2 infection.

Improving uptake of recommended vaccines in medically at-risk children primarily attending hospitals requires an innovative rather than a population approach. Over the past two decades, behavioural scientists have learnt how to design non-coercive 'nudge' strategies to promote positive behaviours in a range of contexts. In psychology and neuroscience, a dual-process theory describes brain functioning as two types of cognitive processes—'system 1' processes described as automatic, uncontrolled, effortless, associative, fast, unconscious and affective, and 'system 2' processes described as reflective, controlled, effortful, rule-based, slow, conscious and rational.[14] Although this 'dual-process' model is considered as a theoretical basis for nudge theory, the nudge approach suggests that automatic decisions can be systematically triggered to improve health outcomes.[15] In other words, nudges are simple cues in our environment that influence people to behave in a certain way to achieve better personal or social goals without making a conscious decision to do so.[16] Using behavioural economics (a discipline that studies how individuals make choice) the environment in which a choice is being made can actively be designed to encourage better health-related choices.[17] A previous study showed nudges and reminders resulted in a decrease in energy-dense nutrient-poor foods in men and sugar-sweetened beverages in women, together with a reduction in body weight.[18] A 'nudge' intervention needs to be simple and low cost and if proven successful can be simply incorporated into standard healthcare. For example, redesigning cardiac rehabilitation referral decision pathway from opt-in to opt-out referral, which automatically identified eligible patients from the electronic health record and notified staff on the wards by using secure text messaging, increased cardiac rehabilitation referral rates from 15% to 85%.[19] Nudge-based interventions to address vaccine hesitancy include using reminders and recall, changing the way information is framed and delivered to an intended audience, changing the messenger delivering information, invoking social norms and emotional affect (eg, through storytelling, dramatic narratives and graphical presentations) and offering incentives or changing defaults. Nudge-based interventions show potential to increase vaccine confidence and uptake. Interestingly, strategies with educational approaches are less effective

(or not effective at all) in improving vaccine uptake.[20] The most promising evidence exists for nudges that offer incentives to parents and healthcare workers, that make information more salient, that frame vaccination as the default or that use trusted messengers to deliver information.[21 22] Normalising vaccination and peer influence can activate social tendencies to join others.[23] Several studies have explored the impact of different interventions on vaccination rates and intentions. A large study with 57 893 participants was conducted in a Northern California health system and found that personal reminder messages increased booster vaccination rates.[24] Among 964 870 participants in 691 820 households in Denmark, two strategies increased influenza vaccination rates.[25 26] In the USA, a study found that short video messages addressing specific COVID-19 vaccine concerns increased vaccination intentions.[27] In a March 2021 study with 1595 participants in Japan, different messages were tested to encourage COVID-19 vaccination. The 'influence-gain' message was effective for older adults. 'Comparison' and 'influence-loss' messages reinforced existing intentions among older adults.[23] In our previous study conducted at a tertiary paediatric hospital in Adelaide (n=600), a significantly greater proportion receiving the SMS intervention were vaccinated with 38.6% in the SMS intervention group compared with 26.2% in the control group.[28] A recent study was conducted in older adult populations (n=48 125) and found behavioural nudges, electronically delivered letters or centralised written reminders, significantly increased influenza vaccination uptake in Finland.[29] A randomised controlled trial (RCT) of a nudge intervention that included text message reminders demonstrated that the first reminder (n=93 354) increased appointment and COVID-19 vaccination rates by 6.07% and 3.57% and that the second reminder (n=67 092) increased those by 1.65% and 1.06% respectively in the early stages of the COVID-19 vaccine rollout.[30] Another RCT was conducted a few months later in a younger population (mean age 39 years; n=142 428) and no SMS message did substantially better or worse than the control whether vaccination rates were measured 1 week after the messages were sent or at the end of the study period.[31] The difference between two studies may suggest that nudges help early in vaccination campaigns, but the efficacy decays. However, nudging-based interventions have shown potential to increase vaccine confidence and uptake in many studies, but further evidence is needed for the development and evaluation of clear recommendations.[22]

Our team recently conducted an RCT from 15 April 15 to 30 September 2021 and found a 47% relative increase in uptake of influenza vaccine in medically at-risk children using a simple SMS nudge codesigned with paediatricians.[28] If our text based nudges with links to short educational videos prove to be successful in improving influenza and COVID-19 vaccine uptake in medically at-risk children, they can easily be implemented as a promising, valuable, low-cost and long-term tool at a national level.

However, a challenge is how behavioural economic concepts and principles can be effectively applied by local community stakeholders to identify barriers and create an innovative intervention to complement the current immunisation programmes. To bridge this gap, a Nudgeathon,[32] a crowdsourcing interaction in which decision groups draw on insights and methods from nudge theory, augmented by design thinking and drama theory to devise implementable solutions to major behavioural policy problems, was launched.[32] These events are typically conducted over 1–2 days, bringing together a diverse range of stakeholders (many who are new to behavioural science) to generate new approaches to complex problems in a time-pressured setting. Three dozen Nudgeathons[32] around the world have been conducted to address multiple issues (eg, handwashing among health providers, increasing regular HIV/sexually transmitted infection testing in men[17 33]) but to our knowledge, there has not been a Nudgeathon for the immunisation programmes. The MINDSPACE framework, which provides a list of behaviour change techniques including Messenger, Incentives, Norms, Defaults, Salience, Priming, Affect, Commitments and Ego, that target the automatic decision processes,[34] was used to identify and generate potential nudges at the Nudgeathon. Our goal through the Nudgeathons is to gain valuable 'customer insight' within the Australian context, fostering a profound understanding of parents' experiences, beliefs, needs and desires, while also identifying the practical and structural challenges they encounter. It's important to recognise that attempts to promote behaviour change without considering these contextual factors often result in frustration. Behaviour change initiatives can be contentious, involving intricate trade-offs and often addressing areas where government decisions are controversial such as COVID-19 vaccination policies. Consequently, innovative approaches like Nudgeathons may be essential to engage the public effectively in exploring acceptable courses of action.

To our knowledge, this will be the first study assessing the effectiveness and cost-effectiveness of a codesigned multicomponent nudge intervention on improving influenza and COVID-19 vaccine uptake in medically at-risk children. Vaccine hesitancy and access barriers are a major barrier that reduces the progress of management of vaccine preventable diseases in children. Identifying solutions to improving COVID-19 and influenza vaccine uptake in medically at-risk children is critical, as these children are more vulnerable to severe COVID-19 and influenza than healthy children.

Therefore, this study aims to design nudge interventions and evaluate the impact of multicomponent nudge interventions on the uptake of influenza and COVID-19 vaccines among medically at-risk children.

## METHODS AND ANALYSIS
Standard Protocol Items: Recommendations for Interventional Trials checklist was used when developing this

protocol.[35] The first participant was randomised on 29 November 2022 and the study is expected to be complete by 31 December 2024. The master protocol was devised to evaluate the impact of multicomponent nudge interventions on the uptake of COVID-19 and influenza vaccines in two distinct subpopulations: medically at-risk children and pregnant women, employing a unified and overarching design. The information pertaining to medically at-risk children is presented here. The protocol information regarding the substudy involving pregnant women was published elsewhere.[36]

## Nudge intervention

In August 2022, a Nudgeathon was held to design appropriate nudges for inclusion in the trial. A total of 14 participants including medically at-risk children, their parents, paediatricians, nurses, hospital administrative personnel, behavioural scientists, psychologists and graphic designers with diverse skills from South Australia, Western Australia and Victoria participated in the Nudgeathon. The participants learnt about behavioural science and successful application of nudges in different disciplines. They were assigned into small groups ensuring that each group comprised participants with different skills. The individual groups were requested to create a nudge for improving either COVID-19 or influenza vaccine uptake in medically at-risk children. At the end of the Nudgeathon each group presented their nudges and one nudge for each condition was chosen by the team to develop the nudge interventions in the following randomised control trials. Some of the most robust effects that have been found to have strong impacts on behaviour[34] such as Messenger, Norms, Salience and Commitments were used to generate nudge interventions in our study. The carefully chosen nudge for both COVID-19 and influenza vaccines consists of three SMS text messages that are sent to parents of medically at-risk children from the participating hospitals 4 weeks apart. The first text message is sent 1 week prior to a scheduled paediatric clinic appointment and provides a link to current national guidelines on COVID-19 or influenza vaccination in children with chronic illness and a link to a video of a paediatrician explaining the benefits of COVID-19 or influenza vaccination in medically at-risk children. The second text message provides a link to a video of a child with chronic illness and their parent giving their opinion on the protection of medically at-risk children against SARS-CoV-2 or influenza infections. The third text message provides a link to a video of a nurse discussing the potential adverse outcomes of not being vaccinated against SARS-CoV-2 or influenza infections. The first and second text messages also provide the option for a parent to agree to vaccinate their children or opt out from receiving further reminders.

The study is aligned with the MINDSPACE framework, a behavioural change model developed by the UK government's Behavioural Insights Team in 2010. The decision to incorporate a blend of behavioural nudges in the intervention stems from the notion that a multifaceted approach can potentially yield a more substantial impact on behaviour. Combining various nudges is often considered a strategy to enhance the likelihood of success, as different individuals may respond diversely to different nudge effects.[37 38] This approach seeks to cast a broader net to appeal to a more extensive spectrum of people and behaviours, leveraging the effects encapsulated in the MINDSPACE. Significantly, there is a considerable overlap among these effects, and the most effective interventions will invariably integrate various elements.

## Primary objectives

► To determine the proportion of medically at-risk children in the intervention versus standard care (non-intervention) group receiving at least one dose of the seasonal influenza vaccine within 3 months after randomisation, as assessed using the Australian Immunisation Register (AIR).

► To determine the proportion of medically at-risk children in the intervention versus standard care (non-intervention) group receiving at least one dose of a COVID-19 vaccine within 3 months after randomisation, as assessed using the AIR.

The AIR is a national register that records vaccines given to all people in Australia including vaccines given under the National Immunisation Programme, including school programmes, and privately, such as for influenza. A recognised vaccination provider including doctors, such as a family physician or community health centre, are required to update the AIR once an influenza, COVID-19 or any vaccine listed in the National Immunisation Programme Schedule is administered.

## Secondary objectives

► To assess the number of medically at-risk children who received COVID-19 or influenza vaccines, change from baseline up to 3 months post randomisation, based on sociodemographic characteristics.

► To assess timeliness of influenza and COVID-19 vaccine uptake among medically at-risk children during the study period by determining the proportion of medically at-risk children who receive the COVID-19 or influenza vaccine by month throughout the study period.

► To estimate the cost-effectiveness of proven interventions compared with standard care in hospital settings.

## Study design and study population

These are parallel-group RCTs (ClinicalTrials.gov Identifier: NCT05613751) designed to measure the impact of a nudge intervention on receipt of one dose of the COVID-19 or influenza vaccine in medically at-risk children who attend a clinic at tertiary hospitals in South Australia, Western Australia and Victoria. Medically at-risk children will be enrolled at seven hospitals across South Australia (Women's and Children's Hospital, Flinders Medical Centre and the Lyell McEwin Hospital), Victoria (The Royal Children's Hospital) and Western Australia

**Table 1**  Inclusion and exclusion criteria

| | Exclusion criteria | Inclusion criteria |
|---|---|---|
| COVID-19 vaccine[41] | ► Known contraindications to COVID-19 vaccine.<br>► Up to date for COVID-19 vaccine (≥two doses) at the time of enrolment.<br>► Sibling of a child already enrolled in the trial (only the sibling who is eligible and scheduled to attend a paediatric clinic first will be eligible).<br>► Previous participation in the influenza nudge RCT. | Medically at-risk children aged 5 years to 18 years with a cardiac, endocrine, respiratory, gastrointestinal, haematological, musculoskeletal, neurological condition. |
| Influenza vaccine[5] | ► Known contraindications to influenza vaccine.<br>► Already received an influenza vaccine during the influenza season in 2023.<br>► Sibling of a child already participating in the trial (the sibling who is eligible and scheduled to attend a paediatric clinic first will be eligible).<br>► Previous participation in the COVID-19 nudge RCT. | Children aged≥6 months and <18 years with medical conditions specified in this list: immunocompromising conditions including malignancy, chronic steroid use, haematopoietic stem cell transplant; functional or anatomical asplenia including sickle cell disease or other haemoglobinopathies, congenital or acquired asplenia (eg, splenectomy) or hyposplenia; cardiac disease including cyanotic congenital heart disease, congestive heart failure, coronary artery disease; chronic respiratory conditions including suppurative lung disease, bronchiectasis, cystic fibrosis, chronic obstructive pulmonary disease, severe asthma (requiring frequent medical consultations or the use of multiple medicines); chronic neurological conditions including hereditary and degenerative CNS diseases, seizure disorders, spinal cord injuries, neuromuscular disorders; chronic metabolic disorders including type 1 or 2 diabetes, amino acid disorders, carbohydrate disorders, cholesterol biosynthesis disorders, fatty acid oxidation defects, lactic acidosis, mitochondrial disorders, organic acid disorders, urea cycle disorders, vitamin/cofactor disorders, porphyria; chronic renal failure; children aged 5–10 years receiving long-term aspiring therapy; Down syndrome; obesity (body mass index $\geq 30 \, kg/m^2$); children born less than 37 weeks gestation. |

CNS, Central Nervous System; RCT, randomised controlled trial.

(Perth Children's Hospital). Each of these hospitals covers different sociodemographic areas and patient sources ensuring our research is generalisable. COVID-19 or influenza vaccination status of medically at-risk children attending the participating hospitals will be determined by checking the AIR. The inclusion and exclusion criteria are listed in table 1.

### Randomisation
Medically at-risk children will be randomised to the intervention and standard care groups in a 1:1 ratio. The randomisation list will be generated using R V.4.02 by an independent statistician who will not be involved with the conduct or analysis of the RCT. Allocations will be performed using randomly permuted blocks, stratified by hospital (ie, separate randomisation sequence used in each hospital). Randomisation will be done on a password protected REDCap database securely held on the University of Adelaide server.

Endpoint measurements will involve low level contact of study staff, however when contact is required (such as AIR/confirmation of influenza/COVID-19 vaccine receipt, etc), this will be carried out by trial staff shielded from information that might reveal trial group assignment. The study statistician undertaking the analysis and study investigators will remain blinded to trial intervention assignment.

### Study processes
Medically at-risk children attending clinics at the hospitals will be identified from Outpatients' Department's appointment lists. Research nurses will assess COVID-19 or influenza vaccination status of children on the AIR to screen for eligibility. Medically at-risk children will be randomised to the intervention arm or the standard care arm on REDCap. Demographic data will be obtained from hospital medical records. Using the Norms, Messenger, Salience and Commitment nudge strategies,

parents of medically at-risk children randomised to the intervention arm will receive a maximum of three SMS reminders from the hospital using 'Message Media' software from the hospitals. The SMS messages will comprise a brief message that many children with chronic illness obtain the COVID-19 or influenza vaccine (Norms) and a reminder for the child to obtain the vaccine. Parents of medically at-risk children have the option of responding to the message by (1) agreeing to obtain the vaccine (Commitment), (2) stating that the participant has already received the vaccine or (3) requesting to opt out from receiving further reminders (first and second SMS messages). Each SMS message also provides a link to a video of (Messenger): (1) a paediatrician discussing the potential serious health consequences of SARS-CoV-2 or influenza infections (first SMS), (2) a medically at-risk child and their parent stating that the child received the vaccine and that the child is protected from serious adverse effects of SARS-CoV-2 or influenza infections (second SMS), (3) a nurse stating the benefits of COVID-19 or influenza vaccination for medically at-risk children (third SMS). In addition, the first SMS also states children with special medical conditions are at increased risk of severe COVID-19 or Influenza disease and provides a link to the current guidelines and safety information about COVID-19 or influenza vaccination (Salience). The paediatrician and nurse assumed the roles of healthcare experts, while the child with medical risks and their parent acted as relatable peers with a similar background. They served as influential messengers in persuading parents to vaccinate their children. The message highlighting the fact that many children with chronic illnesses receive the COVID-19 or influenza vaccine used a social norms approach, conveying what other parents in similar situations typically do. Meanwhile, the message emphasising the increased risk of severe COVID-19 or influenza for children with special medical conditions was designed to capture parents' attention, as it is information more likely to be comprehensible and directly relevant to their own children. The response options were strategically crafted to foster reciprocity and serve as commitments. The first SMS is sent approximately 1 week prior to a scheduled clinic visit. The second SMS message is sent 4 weeks after the first message to those who have not received a dose of COVID-19 or influenza vaccine after the first SMS (confirmed on AIR) and not requested to opt out. The third SMS message is sent 4 weeks after the second message to those who have not received a dose of COVID-19 or influenza vaccine after the second SMS (confirmed on AIR) and not requested to opt out. The AIR will be checked at 3 months after randomisation to assess whether medically at-risk children have received a dose of COVID-19 or influenza vaccine. All data will be stored securely on a password protected REDCap database held by the University of Adelaide. All data will be deidentified prior to presentation and publication.

## Study monitoring and surveillance

The nudge is a behavioural change intervention and hence there are no risks related to invasive procedures or investigational medications. Any risks of mental distress associated with receiving the text message is expected to be very low and the parents have the option of opting out from receiving the further text messages after the first one. However, a risk assessment and management plan has been developed for this study. Moreover, the study management committee comprising the chief investigator, site investigators, study coordinators and statistician will carefully monitor the progress of the study.

This trial has certain limitations. The nudge interventions incorporated a blend of various nudge techniques, and we did not assess the individual effects of these techniques in our study. We included links within the text messages, and it was the parents' responsibility to click on these links. If the links were not clicked, the nudge interventions may not have been as effectively implemented as originally intended.

## Sample size and statistical analysis plan

To detect an increase in the vaccination rate from ~50% in the control group to 60% in the intervention group, which is considered a clinically relevant impact, with 90% power (two-sided test with alpha=0.05), a sample size of n=519 per group (1038 total) is required. Should the vaccination rate in the control arm be higher than 50% or lower than 40%, this sample size will still provide at least 90% power to detect a 10% absolute increase in vaccination with the nudge intervention. We aim to enrol at least 1038 medically at-risk children to each of the two RCTs across the participating hospitals.

Statistical analyses will be conducted on an intention-to-treat basis according to a prespecified statistical analysis plan. Baseline characteristics including age, socioeconomic status based on postcode and ethnicity will be reported for each group using percentages, means with SD or medians with ranges as appropriate. The vaccination rate will be compared between intervention and control groups using logistic regression, with adjustment only made for participating hospitals (treated as a categorical fixed effect). Intervention effects will be reported as ORs with 95% CIs. Patient characteristics associated with vaccination will be investigated using multivariable logistic regression models. Cost-effectiveness analysis of the nudge intervention will be compared with standard care from the healthcare payer perspective. The primary cost-effectiveness analysis will estimate the incremental cost per additional person vaccinated, in each of the two RCTs. Secondary analyses for the trial focused on influenza vaccination will estimate the cost per quality-adjusted life year (QALY) gained. The study results will be used to estimate the impact of the nudge intervention on health-related quality of life by calculating the QALYs gained as a result of the increased vaccine uptake. Implementation costs will be obtained from the study budget and costs related to research activities will be excluded. Estimated

cost offsets to the health system associated with influenza related disease (eg, hospitalisations and emergency visits) will be derived from the literature and calculated using cost weights for Australian Refined Diagnosis Related Groups (AR-DRGs). The cost-effectiveness evaluation will follow standard reporting guidelines in the Consolidated Health Economic Evaluation Reporting Standards (CHEERS) statement.[39]

## Patient and public involvement

In our previous survey,[6] 539 parents were interviewed, revealing that parents of children with chronic medical conditions were less likely to vaccinate their children unless recommended by a specialist. Interventions based on nudging in our previous study[28] showed potential in increasing vaccine uptake. Consequently, the study aims, design and outcome measures have been informed by the priorities, experiences and preferences of parents. Medically at-risk children and their parents actively contributed to the study's design during the Nudgeathon. Their insights played a crucial role in designing the nudge interventions, making it more relevant and patient centred. The study results will be disseminated to the trial participants, which includes medically at-risk children and their parents, through various accessible means, such as academic publication and lay summaries on our research unit's webpage, to ensure that the findings are easily understood and meaningful to this group.

## ETHICS AND DISSEMINATION

The current protocol V.2.0 (05 January 2023) and all study material have been reviewed and approved by the Women's and Children's Health Network Human Research Ethics Committee (HREC/2022/00082) and research governance approval has been obtained from the Women's and Children's Hospital, Flinders Medical Centre and Lyell McEwin Hospital in South Australia, The Royal Children's Hospital in Victoria and Perth Children's Hospital in Western Australia. The written consent was obtained from all stakeholders including health service leaders and clinicians and health consumers to participate in the Nudgeathons. A waiver of consent was approved for medically at-risk children to participate in this trial and receive SMS messages in accordance with the National Statement on Ethical Conduct in Human Research. Three criteria guide this. First, there are no estimated risks or harms associated with study procedures. Any risk of psychosocial distress associated with receiving the nudge interventions is very unlikely. Second, it would not be possible to obtain informed consent without threatening the validity of the trials. It would be impossible to assess the effectiveness of the nudge by recruiting parents of medically at-risk children and informing them about the study and then randomising them not to receive the nudge. Third, as influenza and COVID-19 vaccines are recommended for all medically at-risk children aged ≥6 months, all data for the influenza and COVID-19 RCTs will be collected as part of routine care, including patient demographics and specialties seen. In addition, this methodology was approved and successfully applied in the Flutext-4U study.[40] We identified a significant difference in uptake of influenza vaccine in medically at-risk children in the Flutext-4U study, likely due to this being a population inclusive study unaffected by healthy selection bias.[28] The study is being conducted in accordance with the Declaration of Helsinki and the International Conference on Harmonization Guidelines on Good Clinical Practice. The trial is being conducted in compliance with the current version of the protocol. Any change to the protocol that affects scientific content, study design and participating will be considered an amendment and will be submitted to HREC for review and approval prior to implementation.

Participant confidentiality is strictly held in trust by the investigators. Therefore, study documentation, data, and all other information generated will be held in strict confidence. No information concerning the study, or the data will be released to any unauthorised third party. Study participant research data, which is for purposes of statistical analysis and scientific reporting, will be transmitted to and stored at the Women's and Children's Hospital. At the end of the study, all study databases will be de-identified and archived at the University of Adelaide. After the completion of the study, the de-identified and aggregated results will be presented at scientific forums and submitted for publication in peer-reviewed journals. The results will be disseminated regardless of the direction of effect. The findings of the study will be communicated to key stakeholders.

The burden of the intervention is planned to be assessed with input from medically at-risk children's parents using a short evaluation questionnaire. Special recognition and appreciation were extended to the medical at-risk children and their parents who participated in the Nudgeathon to highlight their valuable contributions to the study's design and success.

## Author affiliations

[1]Vaccinology and Immunology Research Trials Unit, Women's and Children's Hospital, Adelaide, South Australia, Australia
[2]Robinson Research Institute, The University of Adelaide, Adelaide, South Australia, Australia
[3]Adelaide Medical School, The University of Adelaide, Adelaide, South Australia, Australia
[4]Murdoch Childrens Research Institute, Melbourne, Victoria, Australia
[5]The Royal Children's Hospital, Melbourne, Victoria, Australia
[6]Department of Paediatrics, The University of Melbourne, Melbourne, Victoria, Australia
[7]Wesfarmers Centre of Vaccines and Infectious Diseases, Telethon Kids Institute and School of Medicine, University of Western Australia, Perth, Western Australia, Australia
[8]Department of Infectious Diseases, Perth Children's Hospital, Perth, Western Australia, Australia
[9]Department of Microbiology, PathWest Laboratory Medicine, Queen Elizabeth II Medical Centre (QEIIMC), Perth, Western Australia, Australia
[10]School of Business, Warwick University, Warwick, UK
[11]Melbourne Sexual Health Clinic & LSHTM, Monash University, Carlton, Victoria, Australia

[12]Women's and Babies Division, Women's and Children's Hospital, Adelaide, South Australia, Australia
[13]Division of Paediatrics, Women's and Children's Hospital, Adelaide, South Australia, Australia
[14]South Australian Health and Medical Research Institute, Adelaide, South Australia, Australia
[15]School of Public Health, The University of Adelaide, Adelaide, Adelaide, South Australia, Australia
[16]Discipline of Public Health, Flinders University, Adelaide, South Australia, Australia
[17]SA Health, South Australian Government, Adelaide, South Australia, Australia
[18]Discipline of Paediatrics, Flinders University, Adelaide, South Australia, Australia
[19]Discipline of Paediatrics Lyell McEwin Hospital, Elizabeth Vale, South Australia, Australia
[20]Discipline of Women's Health, Lyell McEwin Hospital, Elizabeth Vale, South Australia, Australia
[21]Discipline of Paediatrics, University of Sydney, Sydney, New South Wales, Australia
[22]Children's Hospital Westmead, Sydney, New South Wales, Australia
[23]Discipline of Public Health, Australian National University, Canberra, ACT, Australia

**Acknowledgements** We acknowledge the following study coordinators for their contribution to study set up and recruitment: Kirsty Herewane (Women's and Children's Hospital), Joanne Koch (Lyell McEwin Hospital), Lauren Thompson (Flinders Medical Centre), Ashleigh Rak (Royal Children's Hospital), Deborah Pidd (Mercy Hospital for Women), Louisa Paparo, Rebecca Pavlos and Erin Van Der Helder (Perth Children's Hospital). We would also like to express our gratitude to all participants who were involved in this study.

**Contributors** HM, IV, JO, MD, CCB, BW, PA, JMD, JC, TRS, JK, NS, MC, DM, DS, GD, SC, JT, NW and LJW conceptualised the interventions, contributed to the design of the study protocol and acquired funding. TRS provided the statistical consult. BW drafted the manuscript. MD, CCB, JMD, JC, JK, NS, MC, DM, DS, GD, SC, NW and LJW critically reviewed the manuscript. HM, BW, IV, JO, TRS, JT and PA critically reviewed and edited the manuscript. HM, BW, IV, JO, MD, CCB, PA, JMD, JC, TRS, JK, NS, MC, DM, DS, GD, SC, JT, NW and LJW approved its final version.

**Funding** This work is supported by the National Health and Medical Research Council of Australia (NHMRC Partnership grant APP2014684). Additional in-kind funding is provided by funding partners including the Department of Health, South Australia Government; Department of Health, Victorian Government; Department of Health, Western Australian Government; Women's and Children's Health Network; Southern Adelaide Local Health Network; Northern Adelaide Local Health Network; Women's and Children's Hospital Foundation; Department of Trade and Investment, South Australian Government, in support of the NHMRC Partnership grant.

**Competing interests** HM acknowledges support from the National Health and Medical Research Council of Australia: Practitioner Fellowship (APP1155066). CCB acknowledges support from the National Health and Medical Research Council of Australia: Investigator Grant (APP1173163). HM is an independent investigator on clinical trials of investigational vaccines manufactured by pharmaceutical companies including Pfizer, ILiAD Biotechnologies and Merck. The institution at which HM, BW and PA are employed has received funding for investigator-led research from GlaxoSmithKline, Sanofi-Pasteur and Pfizer Vaccines. All authors receive no personal payments from industry. There are no other conflicts of interest to declare.

**Patient and public involvement** Patients and/or the public were involved in the design, or conduct, or reporting, or dissemination plans of this research. Refer to the Methods and analysis section for further details.

**Patient consent for publication** Not applicable.

**Provenance and peer review** Not commissioned; externally peer reviewed.

**ORCID iDs**
Ivo Vlaev http://orcid.org/0000-0002-3218-0144
Jodie M Dodd http://orcid.org/0000-0002-6363-4874
Thomas R Sullivan http://orcid.org/0000-0002-6930-5406
Jane Tuckerman http://orcid.org/0000-0001-6938-1751
Helen Marshall http://orcid.org/0000-0003-2521-5166

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
