## [Reviewer comments · BMJ Open]

ARTICLE DETAILS

TITLE (PROVISIONAL)	Nudging toward COVID-19 and influenza vaccination uptake in medically at risk children – EPIC study protocol of randomised controlled trials in Australian paediatric outpatient clinics
AUTHORS	Wang, Bing; Andraweera, Prabha; Danchin, Margaret; Blyth, Christopher C.; Vlaev, Ivo; Ong, Jason; Dodd, Jodie; Couper, Jennifer; Sullivan, Thomas; Karnon, Jonathan; Spurrier, Nicola; Cusack, Michael; Mordaunt, Dylan; Simatos, Dimi; Dekker, Gustaaf; Carlson, Samantha; Tuckerman, Jane; Wood, Nicholas; Whop, Lisa; Marshall, Helen

VERSION 1 – REVIEW

REVIEWER	Johansen, Niklas Copenhagen University Hospital, Department of Cardiology, Herlev and Gentofte Hospital
REVIEW RETURNED	06-Jul-2023

GENERAL COMMENTS	The manuscript by Wang et al. describes the study protocols for 2 separate implementation trials in children with chronic medical conditions investigating the effects of nudges to increase influenza and COVID-19 vaccination, respectively. The trials are well-designed and address an important public health topic, and the manuscript is clear and very well written. The waived consent makes perfect sense and substantially strengthens the validity of the trial. I have the following comments: 1: The nudging intervention is meticulously designed, but the way that it is delivered seems to make it difficult to tease out which nudging strategies are actually the most effective. The intervention deploys 4 different nudging strategies simultaneously. Did the authors consider other design features to be able to separate the effects of the different nudging strategies? 2: Could the authors please elaborate on why they only chose to have one intervention arm instead of having several intervention arms, which would allow for testing of additional nudging strategies? 3: Several prior influenza vaccination nudging trials have found text stating that vaccines were "reserved for you" the most effective. Did the authors consider including such a nudge? 4: Do the nudges contain any information on where to obtain vaccination?
---

	5: It seems that participation in the COVID RCT precludes participation in the influenza RCT, but not vice versa. Why? The rationale for this is not currently described in the manuscript. 6: p. 9, line 206: " The first participant was randomised on 29 November 2023 2022" – which is it? 2022 I guess? 7: Ref 32 should be the NUDGE-FLU results paper from The Lancet. The reference currently cited is the design paper which does not contain any results. 8: The manuscript would benefit from the addition of a section discussing the potential limitations of the trials.
--	--

REVIEWER	Thunström, Linda University of Wyoming, Department of Economics
REVIEW RETURNED	05-Oct-2023

GENERAL COMMENTS	Referee report, bmjopen-2023-076194: "Nudging toward COVID-19 and influenza vaccination uptake in medically at risk children – EPIC study protocol of randomised controlled trials in children." This study will use two separate RCTs to examine the impact of a nudge intervention on the uptake of COVID-19 vaccines (RCT 1) and influenza vaccines (RCT 2) in at risk children. The intervention consists of a multitude of nudges, across time, sent to the parents of children at risk. The combination of nudges used in the intervention was determined based on a "nudgeathon," a workshop with stakeholders. The authors aim to recruit N=1,038 to each RCT. The study addresses an important issue and has the potential to generate highly valuable information and outcomes. The study has two major strengths: (1) it tests an intervention aimed to increase the vaccine uptake for a group of people of particular importance (vulnerable children), i.e., it could easily be argued that this specific group is one that we as a society care for the most (see e.g., age adjusted VSL), and, (2) it will generate data on actual outcomes (as opposed to intentions), in a setting (health decisions) where it is notoriously difficult to obtain data on "consumer" choices. While the study protocol overall is clear and easy to follow, it would benefit from clarifications. Please find my comments below, in no particular order of importance. 1. It would be beneficial with more detail on the basis for the intervention. The authors mention that a nudgeathon was used to identify/design the intervention. However, what was the rationale behind designing the intervention to entail a combination of nudges (as opposed to a single nudge)? What was the rationale behind choosing these particular nudges? Further, the study protocol would benefit from describing how and why each of the nudges are expected to affect behavior. There are quite a few studies on the effectiveness of nudges in increasing influenza vaccine uptake (and, more recently, the COVID-19 vaccine uptake). Did the results from the existing literature have any bearing on the choice of nudges in the current study? If so, how? If not, why? It seems the authors would want to design the intervention with the highest potential to affect vaccine uptake, and
---

	it is unclear how that was achieved via the nudgeathon. Why would the outcome of the nudgeathon be expected to be better than simply identifying the most effective nudges based on existing literature on nudges and vaccine uptake, and test whether those same nudges are the most effective also on the population of interest to the current study? 2. Related to the above, it would be beneficial with more detail on the outcome of other RCTs that measure the impact of nudges on vaccine uptake, e.g., the findings from Renosa et al. (2021), which is referenced in the study protocol, as well as other RCTs that have been published since (some of them referenced in the study protocol). Those details would ideally include the type of nudge(s) tested in the study, the study population and the outcome on vaccinations, including the effect size. The latter would be useful to judge whether the statistical power is sufficient in the current protocol. 3. Also related, on page 11, line 225, you mention Messenger, Norm, Salience and Commitments, but exactly how does your intervention address all of these? And what do you mean by “some of the most robust effects?” In what context, and for what population? Is it likely to transfer to your context and population? 4. More information on the randomization would be beneficial. You state that “Allocations will be performed using randomly permuted blocks, stratified by site.” Is a “site” a hospital that participates in the RCT? Will you perform separate randomizations into the intervention at each hospital, or do you lump all hospitals together, and then perform the randomization? 5. It is possible that a meaningful part of the effect from the intervention is speeding up vaccinations that otherwise would still have occurred (just later). How will that impact the outcome of your study, or interpretation of your results? 6. It would be beneficial if more detail was provided on the logistic regression. What will be the specification of the regression model (variables included and their assumed functional relationship with the outcome variable)? How will you deal with the fact that some people might only receive one SMS (if they got vaccinated after the first SMS), while others might receive more than one SMS? Any concerns about data collection potentially stretching across a longer time period, such that conditions (e.g., risks from COVID-19 and the flu) might change? If so, how will you address such concerns in your data analysis or interpretation of your results? Typo: I assume that the year on page 9, line 206, should be 2022.
--	---

VERSION 1 – AUTHOR RESPONSE

Reviewer: 1

Dr. Niklas Johansen, Copenhagen University Hospital

Comments to the Author:

The manuscript by Wang et al. describes the study protocols for 2 separate implementation trials in children with chronic medical conditions investigating the effects of nudges to increase influenza and

COVID-19 vaccination, respectively. The trials are well-designed and address an important public health topic, and the manuscript is clear and very well written. The waived consent makes perfect sense and substantially strengthens the validity of the trial. I have the following comments:

1: The nudging intervention is meticulously designed, but the way that it is delivered seems to make it difficult to tease out which nudging strategies are actually the most effective. The intervention deploys 4 different nudging strategies simultaneously. Did the authors consider other design features to be able to separate the effects of the different nudging strategies?

Response: Thanks for the reviewers' suggestion. This intervention is multifaceted, aligning with the principles outlined in "A new framework for developing and evaluating complex interventions: update of Medical Research Council guidance" (<https://www.bmj.com/content/374/bmj.n2061>). While a qualitative evaluation is typically required to assess the individual impacts of nudges on recipients, we have chosen not to conduct a process evaluation. Moreover, conducting separate trials to test individual nudges would require a much larger sample size than we can currently recruit or afford to achieve the desired statistical power. In our trials, our primary objective is to maximize the effectiveness of the intervention, which was designed with the involvement of consumers and stakeholders to ensure the inclusion of nudge factors considered important by them. In addition, the following information has been added to the Methods section.

The study is aligned with the MINDSPACE Framework, a behavioural change model developed by the UK government's Behavioural Insights Team in 2010. The decision to incorporate a blend of behavioural nudges in the intervention stems from the notion that a multifaceted approach can potentially yield a more substantial impact on behaviour. Combining various nudges is often considered a strategy to enhance the likelihood of success, as different individuals may respond diversely to different nudge effects.^{38,39} This approach seeks to cast a broader net to appeal to a more extensive spectrum of people and behaviours, leveraging the effects encapsulated in the acronym MINDSPACE, which includes Messenger, Incentives, Norms, Defaults, Salience, Priming, Affect, Commitment, and Ego. Significantly, there is a considerable overlap among these effects, and the most effective interventions will invariably integrate various elements.

37. Peter Howley, Neel Ocean, Can nudging only get you so far? Testing for nudge combination effects, *European Review of Agricultural Economics*, Volume 49, Issue 5, December 2022, Pages 1086–1112, <https://doi.org/10.1093/erae/jbab041>

38. Cosic A, Cosic H, Ille S. Can nudges affect students' green behaviour? A field experiment. *Journal of Behavioral Economics for Policy*. 2018;2(1):107-11.

2: Could the authors please elaborate on why they only chose to have one intervention arm instead of having several intervention arms, which would allow for testing of additional nudging strategies?

Response: See the response above please. Furthermore, the power analysis has indicated that we cannot feasibly recruit a sample size large enough to test individual nudges, particularly given that the expected effect sizes are relatively modest. The required sample size and associated costs would present significant barriers.

3: Several prior influenza vaccination nudging trials have found text stating that vaccines were "reserved for you" the most effective. Did the authors consider including such a nudge?

Response: We conducted literature review at the stage of protocol development. However, we took a different approach to develop nudge interventions using Nudgeathons in order to codesign the preferred nudges together with the consumers and other stakeholders. This 'reserved for you' nudge was designed based on the concept of loss aversion within the incentive principle of the MINDSPACE framework. However, during our workshop, the participants did not judge this nudge likely to be

viewed as acceptable, and this decision was based on our adherence to the 'APEASE' criteria (Acceptability, Practicability, Effectiveness, Affordability, Side-effects, and Equity) which have been developed to assist researchers in designing and evaluating behaviour change interventions (https://assets.publishing.service.gov.uk/media/5e7b4e85d3bf7f133c923435/PH_EBI_Achieving_Behaviour_Change_Local_Government.pdf). Therefore, the message implying that vaccines were "reserved for you" was not considered suitable.

4: Do the nudges contain any information on where to obtain vaccination?

Response: No, this is a nationwide multicenter study. Vaccination providers are situated differently across the various centers. Potential nudges including any information on where to access vaccination were also not raised in the Nudgeathon. Consequently, we did not include information about where to access vaccinations.

5: It seems that participation in the COVID RCT precludes participation in the influenza RCT, but not vice versa. Why? The rationale for this is not currently described in the manuscript.

Response: Australia is located in the Southern Hemisphere, where the seasons are opposite to those in European countries. In Australia, the flu season typically starts in March-April, and our recruitment began in October 2022. This is why we chose to initiate the COVID RCT first. Participants were limited to participating in a single trial, with the COVID-19 RCT concluding first. However, there was a slight overlap in the recruitment periods for the COVID-19 and influenza RCTs. To prevent any potential confusion, the following information has been added to the inclusion/exclusion criteria.

Exclusion criteria

- *Previous participation in the influenza nudge RCT*

6: p. 9, line 206: " The first participant was randomised on 29 November 2023 2022" – which is it? 2022 I guess?

Response: Thanks. Yes, 2022. The error has been corrected in the manuscript.

7: Ref 32 should be the NUDGE-FLU results paper from The Lancet. The reference currently cited is the design paper which does not contain any results.

Response: Thanks, Niklas. The Lancet paper has been cited too.

8: The manuscript would benefit from the addition of a section discussing the potential limitations of the trials.

Response: the following information has been added to the method section.

This trial has certain limitations. The nudge interventions incorporated a blend of various nudge techniques, and we did not assess the individual effects of these techniques in our study. We included links within the text messages, and it was the parents' responsibility to click on these links. If the links were not clicked, the nudge interventions may not have been as effectively implemented as originally intended.

Reviewer: 2

Linda Thunström, University of Wyoming

Comments to the Author:

Referee report, bmjopen-2023-076194: "Nudging toward COVID-19 and influenza vaccination uptake in medically at risk children – EPIC study protocol of randomised controlled trials in children."

This study will use two separate RCTs to examine the impact of a nudge intervention on the uptake of COVID-19 vaccines (RCT 1) and influenza vaccines (RCT 2) in at risk children. The intervention consists of a multitude of nudges, across time, sent to the parents of children at risk. The combination

of nudges used in the intervention was determined based on a “nudgeathon,” a workshop with stakeholders. The authors aim to recruit N=1,038 to each RCT.

The study addresses an important issue and has the potential to generate highly valuable information and outcomes. The study has two major strengths: (1) it tests an intervention aimed to increase the vaccine uptake for a group of people of particular importance (vulnerable children), i.e., it could easily be argued that this specific group is one that we as a society care for the most (see e.g., age adjusted VSL), and, (2) it will generate data on actual outcomes (as opposed to intentions), in a setting (health decisions) where it is notoriously difficult to obtain data on “consumer” choices.

While the study protocol overall is clear and easy to follow, it would benefit from clarifications. Please find my comments below, in no particular order of importance.

1. It would be beneficial with more detail on the basis for the intervention. The authors mention that a nudgeathon was used to identify/design the intervention. However, what was the rationale behind designing the intervention to entail a combination of nudges (as opposed to a single nudge)? What was the rationale behind choosing these particular nudges? Further, the study protocol would benefit from describing how and why each of the nudges are expected to affect behavior. There are quite a few studies on the effectiveness of nudges in increasing influenza vaccine uptake (and, more recently, the COVID-19 vaccine uptake). Did the results from the existing literature have any bearing on the choice of nudges in the current study? If so, how? If not, why? It seems the authors would want to design the intervention with the highest potential to affect vaccine uptake, and it is unclear how that was achieved via the nudgeathon. Why would the outcome of the nudgeathon be expected to be better than simply identifying the most effective nudges based on existing literature on nudges and vaccine uptake, and test whether those same nudges are the most effective also on the population of interest to the current study?

Response: Nudges are highly contextual, meaning that a nudge that works effectively in one setting may not be equally successful in another. Our approach involved the use of Nudgeathons, where the intention was to collaboratively design nudges with the consumers and stakeholders. Each team participating in the Nudgeathon was provided with insights into various effective nudging techniques, including examples of strategies to boost vaccine uptake. During the protocol development phase, we conducted a thorough literature review, drawing on insights from our prior research and existing scholarly works. These findings heavily influenced the design of the nudge interventions in the present study. Moreover, our approach is rooted in a bottom-up design philosophy, relying solely on input from consumers, stakeholders and experts. This approach is distinct and aligns more closely with the principles of 'design thinking' than the conventional methodologies used for designing healthcare interventions. The development of nudges was also guided by the 'APEASE' criteria (Acceptability, Practicability, Effectiveness, Affordability, Side-effects, and Equity), a framework designed to assist researchers in creating and evaluating behaviour change interventions (source: https://assets.publishing.service.gov.uk/media/5e7b4e85d3bf7f133c923435/PHEBI_Achieving_Behaviour_Change_Local_Government.pdf).

The following information has been added to the introduction to provide a clearer rationale for the utilisation of Nudgeathons in the development of nudge interventions, as opposed to relying solely on existing literature.

Our goal through the Nudgeathons is to gain valuable "Customer Insight" within the Australian context, fostering a profound understanding of parents' experiences, beliefs, needs, and desires, while also identifying the practical and structural challenges they encounter. It's important to recognise that attempts to promote behaviour change without considering these contextual factors often result in frustration. Behaviour change initiatives can be contentious, involving intricate trade-offs and often addressing areas where government decisions are controversial such as COVID-19

vaccination policies. Consequently, innovative approaches like Nudgeathons may be essential to engage the public effectively in exploring acceptable courses of action.

2. Related to the above, it would be beneficial with more detail on the outcome of other RCTs that measure the impact of nudges on vaccine uptake, e.g., the findings from Renosa et al. (2021), which is referenced in the study protocol, as well as other RCTs that have been published since (some of them referenced in the study protocol). Those details would ideally include the type of nudge(s) tested in the study, the study population and the outcome on vaccinations, including the effect size. The latter would be useful to judge whether the statistical power is sufficient in the current protocol.

Response: Thanks for the reviewer's suggestion. We chose our sample size according to the smallest difference in vaccine uptake rates between groups that we thought would be clinically important (10% absolute risk difference), rather than an expected difference. A 10% difference does however seem feasible based on previous trials involving nudges for vaccine uptake. To provide additional context regarding previous vaccine uptake studies, we have added the following information (underlined) to the introduction.

Several studies have explored the impact of different interventions on vaccination rates and intentions. A large study with 57,893 participants was conducted in a Northern California health system, and found that personal reminder messages increased booster vaccination rates.²⁴ Among 964,870 participants in 691,820 households in Denmark, two strategies increased influenza vaccination rates.^{25,26} In the US, a study found that short video messages addressing specific COVID-19 vaccine concerns increased vaccination intentions.²⁷ In a March 2021 study with 1,595 participants in Japan, different messages were tested to encourage COVID-19 vaccination. The 'influence-gain' message was effective for older adults. 'Comparison' and 'influence-loss' messages reinforced existing intentions among older adults.²³ In our previous study conducted at a tertiary paediatric hospital in Adelaide (n=600), a significantly greater proportion receiving the SMS intervention were vaccinated with 38.6% in the SMS intervention group compared with 26.2% in the control group.²⁸ A recent study was conducted in older adult populations (n=48,125) and found behavioural nudges, electronically delivered letters or centralized written reminders, significantly increased influenza vaccination uptake in Finland.²⁹ A randomized controlled trial (RCT) of a nudge intervention that included text message reminders demonstrated that the first reminder (n=93,354) increased appointment and COVID-19 vaccination rates by 6.07% and 3.57% and that the second reminder (n=67,092) increased those by 1.65% and 1.06% respectively in the early stages of the COVID-19 vaccine rollout.³⁰ Another RCT was conducted a few months later in a younger population (mean age 39 years; n = 142,428) and no SMS message did substantially better or worse than the control whether vaccination rates were measured one week after the messages were sent or at the end of the study period.³¹ The difference between two studies may suggest that nudges help early in vaccination campaigns, but the efficacy decays. However, nudging-based interventions have shown potential to increase vaccine confidence and uptake in many studies, but further evidence is needed for the development and evaluation of clear recommendations.²²

3. Also related, on page 11, line 225, you mention Messenger, Norm, Salience and Commitments, but exactly how does your intervention address all of these? And what do you mean by "some of the most robust effects?" In what context, and for what population? Is it likely to transfer to your context and population?

Response: The MINDSPACE framework organises nine behavioural science principles that can be used to guide policy design: Messenger, Incentives, Norms, Defaults, Salience, Priming, Affect, Commitments, and Ego. Our nudge interventions used four of those nine principles - Messenger, Norm, Salience and Commitments.

Messenger-we are heavily influenced by who communicates information;

Norms we are strongly influenced by what others do;

Salience our attention is drawn to what is novel and seems relevant to us;
Commitments we seek to be consistent with our public promises, and reciprocate acts;

The following information has been added to further explain how those effects were used in our nudge interventions.

The paediatrician and nurse assumed the roles of healthcare experts, while the child with medical risks and their parent acted as relatable peers with a similar background. They served as influential messengers in persuading parents to vaccinate their children. The message highlighting the fact that many children with chronic illnesses receive the COVID-19 or influenza vaccine utilised a social norms approach, conveying what other parents in similar situations typically do. Meanwhile, the message emphasising the increased risk of severe COVID-19 or influenza for children with special medical conditions was designed to capture parents' attention, as it is information more likely to be comprehensible and directly relevant to their own children. The response options were strategically crafted to foster reciprocity and serve as commitments.

4. More information on the randomization would be beneficial. You state that “Allocations will be performed using randomly permuted blocks, stratified by site.” Is a “site” a hospital that participates in the RCT? Will you perform separate randomizations into the intervention at each hospital, or do you lump all hospitals together, and then perform the randomization?

Response: We have now clarified that randomization is stratified by hospital. The following information has been added to the Methods section.

Allocations will be performed using randomly permuted blocks, stratified by site/hospital (i.e., separate randomisation sequence used in each hospital).

5. It is possible that a meaningful part of the effect from the intervention is speeding up vaccinations that otherwise would still have occurred (just later). How will that impact the outcome of your study, or interpretation of your results?

Response: To account for fluctuations in vaccination rates in the general population after the lifting of COVID-19 restrictions, randomisation will enable us to evaluate the effectiveness of nudge interventions, even in the context of increasing vaccination rates.

6. It would be beneficial if more detail was provided on the logistic regression. What will be the specification of the regression model (variables included and their assumed functional relationship with the outcome variable)? How will you deal with the fact that some people might only receive one SMS (if they got vaccinated after the first SMS), while others might receive more than one SMS? Any concerns about data collection potentially stretching across a longer time period, such that conditions (e.g., risks from COVID-19 and the flu) might change? If so, how will you address such concerns in your data analysis or interpretation of your results?

Response: We appreciate the reviewer's suggestions. We have now clarified that the logistic regression model for the primary outcome will only include adjustment for hospital, treated as a categorical fixed effect. As described in the statistical methods section, analyses will be performed on an intention to treat basis; we do not intend to describe per-protocol effects or explore the effects of changes in COVID-19/flu risk over time, albeit these effects could be investigated in later exploratory analyses. The following sentence has been revised.

The vaccination rate will be compared between intervention and control groups using logistic regression, with adjustment only made for participating hospitals (treated as a categorical fixed effect).

Typo: I assume that the year on page 9, line 206, should be 2022.

Response: Thanks. The typo has been corrected.

Professor Helen Siobhan Marshall AM
MBBS MD MPH DCH FPHAA FAHMS

Clinical Research Director, Women's and Children's Health Network
Consultant in Vaccinology & Medical Director, Vaccinology and Immunology Research Trials Unit,
Women's and Children's Health Network
NHMRC Practitioner Fellow and Professor in Vaccinology, Adelaide Medical School, Deputy Director,
Robinson Research Institute, The University of Adelaide
2022 SA Australian of the Year and Woman of the Year

Kaurna Country, Women's and Children's Hospital, North Adelaide, 5006, SA
T: 8161 8115 Fax: 8161 7031
E: helen.marshall@adelaide.edu.au
W: adelaide.edu.au/robinson-research-institute
Bio: www.adelaide.edu.au/directory/helen.marshall

Page 1 of 10

VERSION 2 – REVIEW

REVIEWER	Johansen, Niklas Copenhagen University Hospital, Department of Cardiology, Herlev and Gentofte Hospital
REVIEW RETURNED	10-Nov-2023
GENERAL COMMENTS	I believe that the authors have sufficiently addressed the comments made by the reviewers and recommend acceptance.